# Reversing sintering effect of Ni particles on γ-Mo₂N via strong metal support interaction

Lili Lin [1,2,11], Jinjia Liu[3,4,11], Xi Liu [5✉], Zirui Gao[2], Ning Rui [6], Siyu Yao[7], Feng Zhang[8], Maolin Wang[2], Chang Liu [9], Lili Han[6], Feng Yang [10], Sen Zhang[9], Xiao-dong Wen [3,4], Sanjaya D. Senanayake[6], Yichao Wu[1], Xiaonian Li[1], José A. Rodriguez [6,8✉] & Ding Ma [2✉]

Reversing the thermal induced sintering phenomenon and forming high temperature stable fine dispersed metallic centers with unique structural and electronic properties is one of the ever-lasting targets of heterogeneous catalysis. Here we report that the dispersion of metallic Ni particles into under-coordinated two-dimensional Ni clusters over γ-Mo₂N is a thermodynamically favorable process based on the AIMD simulation. A Ni-4nm/γ-Mo₂N model catalyst is synthesized and used to further study the reverse sintering effect by the combination of multiple in-situ characterization methods, including in-situ quick XANES and EXAFS, ambient pressure XPS and environmental SE/STEM etc. The under-coordinated two-dimensional layered Ni clusters on molybdenum nitride support generated from the Ni-4nm/γ-Mo₂N has been demonstrated to be a thermally stable catalyst in 50 h stability test in CO₂ hydrogenation, and exhibits a remarkable catalytic selectivity reverse compared with traditional Ni particles-based catalyst, leading to a chemo-specific CO₂ hydrogenation to CO.

[1] Institute of Industrial Catalysis, State Key Laboratory of Green Chemistry Synthesis Technology, College of Chemical Engineering, Zhejiang University of Technology, 310014 Hangzhou, Zhejiang, China. [2] Beijing National Laboratory for Molecular Sciences, College of Chemistry and Molecular Engineering and College of Engineering and BIC-ESAT Peking University, 100871 Beijing, P. R. China. [3] State Key Laboratory of Coal Conversion, Institute of Coal Chemistry, Chinese Academy of Sciences, Taiyuan, China. [4] National Energy Centre for Coal to Liquids, Synfuels China Co. Ltd, Beijing, China. [5] School of Chemistry and Chemical Engineering, In-situ Center for Physical Science, Shanghai Jiao Tong University, Shanghai, China. [6] Chemistry Division, Brookhaven National Laboratory, Upton, NY 11973, USA. [7] Key Laboratory of Biomass Chemical Engineering of Ministry of Education, College of Chemical and Biological Engineering, Zhejiang University, 310027 Hangzhou, China. [8] Materials Science and Chemical Engineering Department, State University of New York, Stony Brook, NY, USA. [9] Department of Chemistry, University of Virginia, Charlottesville, VA 22904, USA. [10] Department of Chemistry, Southern University of Science and Technology, 518055 Shenzhen, China. [11] These authors contributed equally: Lili Lin, Jinjia Liu. ✉email: liuxi@sjtu.edu.cn; rodrigez@bnl.gov; dma@pku.edu.cn

Sintering, the term referring to the high-temperature agglomeration of the fine-dispersed metal species in a heterogeneous catalyst[1–4], is a common phenomenon and one of the major reasons accounting for the deactivation of the working heterogeneous catalysts[5,6]. From thermodynamic point of view, the surface free energy of the nanosized metal clusters rises significantly with the decreasing particle size, due to the enlarged ratio of the under-coordinated surface atoms in the entire particle[7]. As long as the thermal perturbation overcomes the adhesion barrier of the supported particles on the substrate, the highly mobile surface metal particles will tend to agglomerate to reduce the exposing metal surfaces and form larger particles[6,8].

To enhance the sintering-resistance ability of catalysts, a number of confinement strategies that encapsulate the metal particles within the pores/tunnels of the porous materials, such as carbon nanotubes[9,10], zeolites[11,12], and metal-organic frameworks[13–15] (MOFs) etc., have been proposed. The feasibilities of these confinement methods have been widely demonstrated to prevent the migration and interparticle aggregation of metal particles. However, the undesirable activity loss and selectivity change caused by the mass transport limitation of the reactants and/or products are difficult to prevent in the confinement system. Therefore, the concept of reversing the sintering of metal nanoparticles[16–18] which refers to the transformation of large particles with minimized exposing atoms into smaller ones highly exposed to the environment at high temperature and remain stable under reaction condition, appears to be another effective solution. In a recent study, a nitrogen doped carbon (CN) material derived from a MOF material was taken as the anchoring substrate and exhibited a strong capture ability to grasp the Pd atoms from 2 to 3 nm Pd-NPs. Under the strong coordinative interaction of the $CN_x$ site with the Pd atoms, the gradual transformation of the Pd-NPs into single atoms at high temperature was observed[17]. The reverse sintering transformation from nanoparticles to single atoms was also confirmed in the systems of silver particles supported on Hollandite-type manganese oxide and Pt particles loaded on $Fe_2O_3$[19–21]. DFT calculations suggested that the reverse sintering effect requires the coordination of the metal-support bond significantly exceeded the internal metal-metal binding inside the particles, leading to a thermodynamically favorable redispersion of metal particles to the most stable single atomic dispersion geometry.

Downsizing nanoparticles to cluster or even single atoms are highly desirable to maximize the metal utilization, and render the electronic properties of low-coordinated metals[7,22–24]. In addition, the reverse sintering heterogeneous catalysts could even achieve extraordinary catalytic stability under the tough reaction conditions[17,25]. The main issue is the search for suitable hosting materials that possess stronger interaction with the active metal atoms than the metal lattice. Our expertize in molybdenum carbides and nitrides catalysts discloses the novelty of the molybdenum compounds as supports to a variety of metal species[26,27]. The strong interaction between the supports and atomically dispersed species not only modulates atomic configurations of supported species, but also greatly modifies their catalytic activity. It inspires us to develop a universal method to design novel catalytic system via a controllable reverse sintering process. Herein, we present that the pre-synthesized 4 nm Ni nanoparticles loaded on the $\gamma$-$Mo_2N$ were able to transform into under-coordinated rafted-like Ni clusters under the reductive thermal treatment. Static Density functional theory (DFT) calculations and the ab initio molecular dynamic (AIMD) simulations predict that the metallic Ni particles can be dispersed spontaneously on the surface of reduced $\gamma$-$Mo_2N$ under the strong interaction between the Ni and $\gamma$-$Mo_2N$. In contrast, neither the NiO particles nor the Ni on the oxygen terminated $\gamma$-

$Mo_2N$ substrate was able to reversely sinter in the simulation. The redispersion processes were successfully monitored and confirmed by the in-situ characterizations of X-ray absorption fine structure spectroscopy (XAFS), ambient pressure X-ray photo electron spectroscopy (AP-XPS), and the environmental secondary electron/scanning transmission electron microscopy (ESE/STEM). The spontaneous dispersion of Ni nanoparticles to raft-like clusters also tuned the electronic properties of Ni species, endowing the Ni/$\gamma$-$Mo_2N$ exceptional selectivity to CO and an excellent catalytic stability in the high temperature reaction.

## Results

**AIMD simulations of Ni NPs structure evolution over $\gamma$-$Mo_2N$.** In the previous studies, we have discovered the existence of strong metal support interaction between metals and the transition metal carbides and nitrides[28–30]. In all the studied systems, the catalysts exhibited several common properties, including the electron deficient supported metal centers (charge transfer from metal to support) and the capability of maintaining the fine dispersion of metal centers even after high temperature treatment in the activation atmosphere. Evoked by the SMSI effect in the TMCs/TMNs supported catalysts[31], we focused our notice in verifying whether the molybdenum nitride as the alternative host material can spit metals from 3D nanoparticles to sub nanometer or 2D layers, even to single atoms, which is called the reversing sintering effect[17]. The AIMD simulation was applied to investigate the structural evolution of a Ni nanoparticle on the $\gamma$-$Mo_2N$ support under thermal perturbation (Fig. 1 and Supplementary Movie 1–8). The temperature factor was set at 590 °C which is in accordance with the commonly used activation temperature of traditional M/$\gamma$-$Mo_2N$(C) catalysts[28,32]. The initial structure of the Ni/$\gamma$-$Mo_2N$ model at 0 ps was constructed by 19-atom Ni particle placed on a Mo terminated $\gamma$-$Mo_2N$(111) surface, and after 30 ps relaxation the mean potential energies only slightly fluctuated, indicating that the energy and the structure were converged after 30 ps simulations (Supplementary Fig. 1). Due to the strong binding of the Ni with the $\gamma$-$Mo_2N$ substrate, an instant collapse of the 3D structure of Ni particles was observed. Within 10 ps, the Ni atoms have spread into a raft-like monolayer on the $\gamma$-$Mo_2N$ (Fig. 1a). The relative interface area of $Ni_{19}$/$\gamma$-$Mo_2N$(111) (Supplementary Fig. 2), increased dramatically to 4.5 times referring to the initial structure, indicating the dispersion of Ni on the $\gamma$-$Mo_2N$(111) support. No significant structure changes of the raft like particles occurred when the simulation time prolonged to 30 ps and the relative interface area was only slightly increased to 5 (Fig. 1d). In comparison, when the surface of molybdenum nitride was covered with oxygen, the NiO species were found to be incapable to spread into under-coordinated species and the relative interface area of $Ni_{19}O_{19}$/$\gamma$-$Mo_2N$(111) from initial structure to 30 ps kept ~1 (Fig. 1b, d), indicating the strong interaction between Ni and molybdenum nitride requires both materials under reduced state. Meanwhile, no obvious wetting phenomenon is observed on the Ni/$CeO_2$ system (Fig. 1c), except for a small shape reconstruction from the cubic into hemispheric particles. A slightly interface area increasement of $Ni_{19}$/$CeO_2$(111) was observed, which fluctuated ~1.5 (Fig. 1d). It can be seen from Fig. 1e that the relative energy change of the redispersion process of Ni on the $\gamma$-$Mo_2N$ is ~−9.0 eV. In contrast, the $\Delta E$ of the re-shape procedure in the Ni/$CeO_2$ model is much smaller. Additionally, a $Ni_{55}$ particle supported on a $8 \times 8$ supercell of $Mo_2N$(111) slab is also simulated for 30 ps (Supplementary Movie 7–8). It could be found that the configuration change of $Ni_{55}$ on $Mo_2N$(111) is similar to $Ni_{19}$ particle, the raft-like configuration is more favorable based on the energy profiles (Supplementary Fig. 3). This result suggests that the

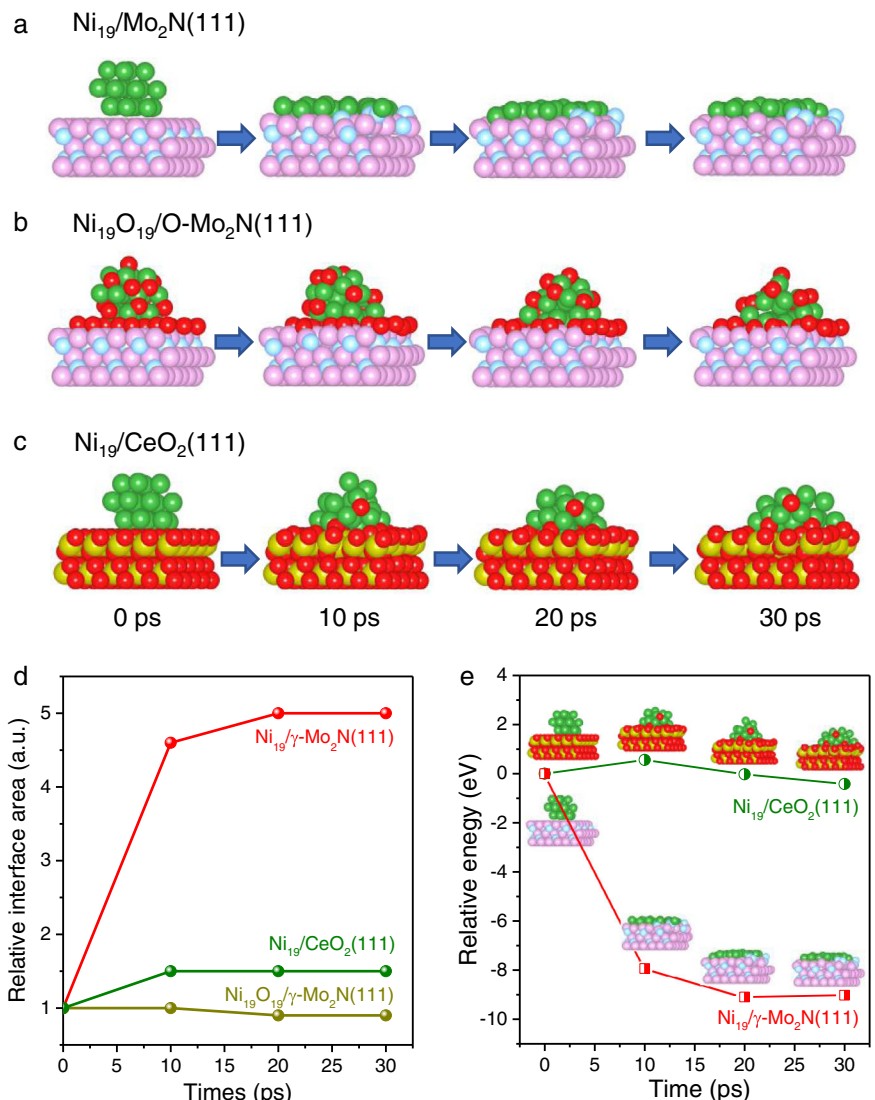

**Fig. 1 The structure evolution of Ni NPs on varied supports at 590 °C in AIMD simulations.** The structure evolution of **a** $Ni_{19}$ on $\gamma$-$Mo_2N$(111) support, **b** $Ni_{19}O_{19}$ on passivated $\gamma$-$Mo_2N$ (O-$Mo_2N$(111)), **c** $Ni_{19}$ on $CeO_2$ support, **d** the relative interface area of Ni to surface atom of support as a function to time, $Ni_{19}$/$\gamma$-$Mo_2N$(111), $Ni_{19}$/$CeO_2$(111) and $Ni_{19}O_{19}$/$\gamma$-$Mo_2N$(111) listed, the initial interface area of the models were taken as bench mark "1". **e** the relative free energy change in the redispersion process of $Ni_{19}$/$\gamma$-$Mo_2N$(111) and $Ni_{19}$/$CeO_2$(111) in 30 ps. In this figure, the red, green, pink, light blue and golden atoms are O, Ni, Mo, N and Ce respectively.

morphology evolution trend is regardless to the size of supported Ni particles. The significant difference of the structure evolution of Ni particles on nitride and oxide supports is controlled by the thermodynamics of the Ni-$\gamma$-$Mo_2N$ and the Ni-$CeO_2$ interfaces.

**Observation of reverse sintering on Ni/$\gamma$-$Mo_2N$ model catalyst.** In order to test the theoretical prediction of AIMD results, a model catalyst composed with the pre-synthesized Ni nanoparticles (4 nm)[33] and $\gamma$-$Mo_2N$ was prepared. The Ni particles were synthesized using the high temperature liquid phase synthesis method with oleylamine and tributylphosphine as the capping reagents[33]. The loading of Ni on the support is controlled at ~2 wt%, and further confirmed by ICP-OES. The TEM image of the as-synthesized Ni NPs demonstrated that the average size is ~4.0 nm with a narrow size distribution and sphere-like shape (Fig. 2a), but the Ni NPs were partially oxidized after deposited onto the passivated FCC structured $\gamma$-$Mo_2N$ support (Fig. 2b). The HAADF-STEM images of the fresh 2%Ni-4 nm/$\gamma$-$Mo_2N$ catalyst were also collected along with the EDX elemental

mapping. It can be seen from the STEM and elemental mapping images that the supported Ni appeared at the surface of $\gamma$-$Mo_2N$ and maintained their original size and shape (Fig. 2c, d).

The prepared model catalysts were used to track the thermal influences on the morphology of the 4 nm supported Ni particles using combination studies of a various of in-situ characterization methods. Both Ni-4nm/$\gamma$-$Mo_2N$ and Ni-4nm/$CeO_2$ catalysts were treated under the reductive atmosphere during a temperature-programmed heating experiment from room temperature to 590 °C. The spectroscopic studies of in-situ quick X-ray adsorption near edge spectroscopy (QXANES) and the ambient pressure X-ray photoelectron spectroscopy (AP-XPS) were performed at the Ni K edge or Ni 3$d$ region to confirm the chemical state and the electronic structure of the supported Ni species (Fig. 3). When the sample was treated in the flow of $N_2$-$H_2$ mixture, a gradual reduction of the sample started at ~360 °C based on the QXANES measurement (Fig. 3a, left panel). The sample was fully reduced to Ni (0) state at ~480 °C judging from the edge position and "white-line" profiles. With the further thermal treatment, an unusual intensity increasing emerged at the

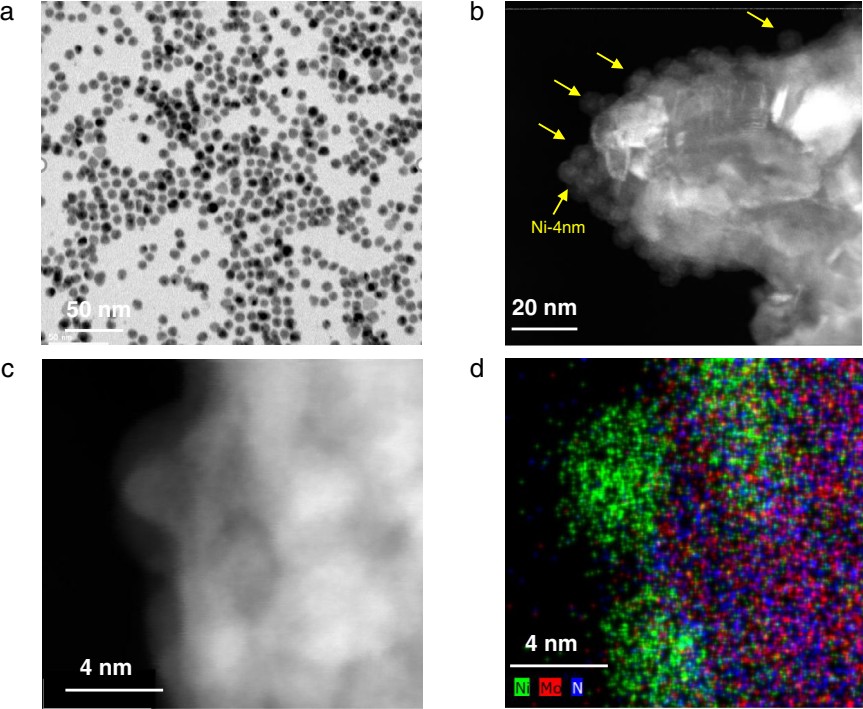

**Fig. 2 The electron microscope images of the fresh Ni-4nm/γ-Mo₂N catalyst. a** TEM image of the Ni-4nm suspension precursor; **b**, **c** STEM images of fresh Ni-4nm/γ-Mo₂N catalyst; **d** The EDS element mapping of fresh Ni-4nm/γ-Mo₂N catalyst, Ni green, Mo red, N blue.

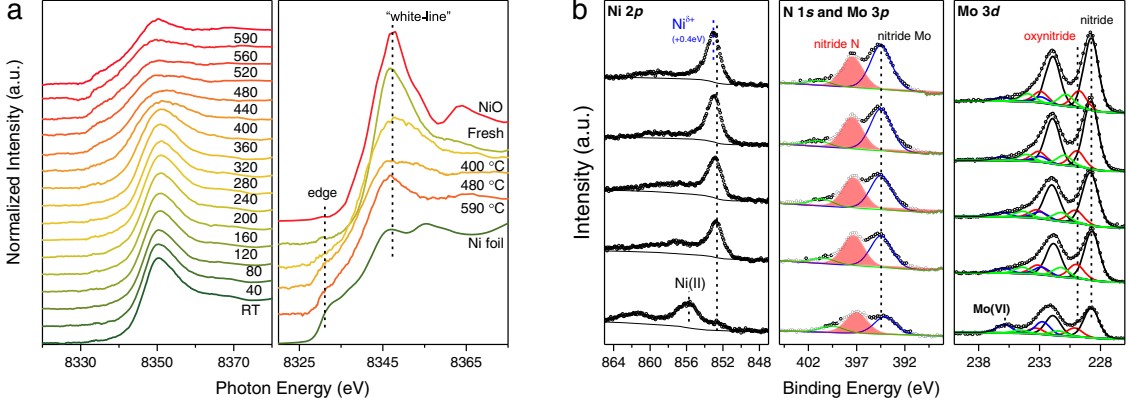

**Fig. 3 In-situ QXANES characterization and AP-XPS spectroscopy of 4%Ni-4nm/γ-Mo₂N catalyst treated at different temperature. a** The QXANES spectra of 4%Ni-4nm/γ-Mo₂N catalyst from room temperature to 590 °C, the right panel of **a** is a comparison of Ni K edge of 4%Ni-4nm/γ-Mo₂N catalyst at state of fresh, reduced at 400 °C, 480 °C, and 590 °C to Ni foil and NiO. **b** AP-XPS profiles of Ni 2*p*, N 1*s* + Mo 3*p* and Mo 3*d* of 4%Ni-4nm/γ-Mo₂N catalyst. The fresh sample was activated in the AP-XPS chamber by a 40 mTorr N₂/H₂ (1:3 v/v) mixture at 300, 350, 400, 500, and 520 °C for 1 h. The spectra at the Ni 2*p*, Mo 3*d*/3*p*, N 1*s*, and C 1*s* XPS regions were collected at each temperature after an hour treatment. The C 1*s* photoemission line with the surface carbon feature (284.8 eV) was used for the binding energy calibration.

"white-line" of Ni-4nm/γ-Mo₂N-590 catalyst (Fig. 3a, right panel), which was probably due to the electron synergistic effect. The higher intensity of the Ni K edge "white-line" signal indicated that the electron density at the Ni site was weakened due to the charge redistribution from Ni to the molybdenum nitride support[28]. The electron structure change of the Ni-4nm/γ-Mo₂N catalysts was further monitored by the surface sensitive AP-XPS technique (Fig. 3b). In the fresh sample, an obvious oxidation layer can be observed at the γ-Mo₂N surface. The Mo (V) and Mo (VI) species were ~31% of the total Mo species. Meanwhile, most of the Ni species was NiO and only a small amount of Ni(0) species could be seen. After the activation, the signals of molybdenum nitride increased and the reduction of the supported Ni occurred. With the increasing temperature, the

Ni(0) binding energy of Ni-4nm/γ-Mo₂N-520 shifted ~0.4 eV positively[28], which is probably related to the charge transfer from Ni species to the γ-Mo₂N supports, in good agreement with the changing of "white line" of QXANES characterization. The N 1*s* XPS peak overlaps with the Mo 3*p*3/2 peaks, making it hard to determine the detailed change in spectra. However, it could still be observed that the major N species on the surface of catalyst is the nitride N centered at 398.3 eV, which maintained its intensity during from 350 to 520 °C.

Furthermore, in-situ X-ray adsorption fine structure[34] (XAFS) of Ni K edge was carried out to evaluate the size of Ni domains and the surrounding coordination environment changes after the thermal treatments at 400 and 590 °C for 40 min. Figure 4a presents the Ni K edge XANES spectra of Ni-4nm/γ-Mo₂N and

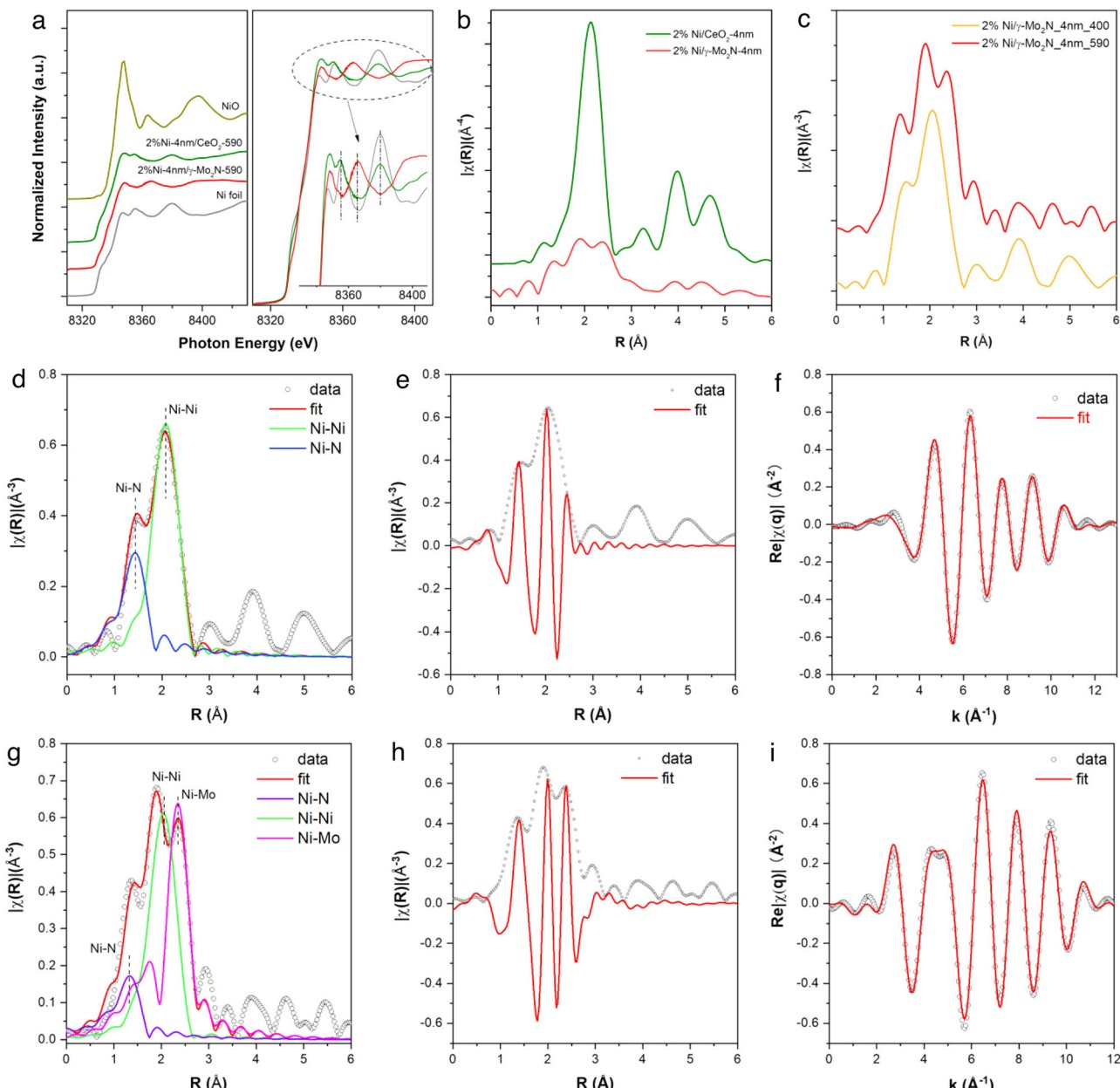

**Fig. 4 In-situ XAFS characterizations of 2%Ni-4nm/γ-Mo₂N catalyst treated at different temperatures of 400 and 590 °C in the atmosphere of H₂/N₂ at a ratio of 3:1. a** The evolution of Ni feature of Ni-4nm/γ-Mo₂N and Ni-4nm/CeO₂ from Ni K edge XANES, and the detailed comparison of XANES is presented at the right panel; The EXAFS figures at R space of **b** 2%Ni-4nm/γ-Mo₂N-590 and 2%Ni-4nm/CeO₂-590, and **c** 2%Ni-4nm/γ-Mo₂N-400 and 2%Ni-4nm/γ-Mo₂N-590; fitting details for Ni K-edge EXAFS spectra obtained for **d-f** 2%Ni-4nm/γ-Mo₂N-400, and **g-i** 2%Ni-4nm/γ-Mo₂N-590.

Ni-4nm/CeO₂ catalysts after 590 °C reduction. Compared with the Ni and NiO standards, the Ni-4nm/CeO₂-590 catalysts exhibited similar pre-edge and near edge features with the Ni foil, indicating the supported Ni species were almost fully reduced after the high temperature reduction. In comparison, the pre-edge feature of Ni/γ-Mo₂N-590 catalyst is slightly weaker than the metallic standard and the XANES oscillation appeared at 8365 eV and higher energy regions cannot be described by neither the Ni (0) nor the NiO (Fig. 4a, the right panel). This phenomenon suggested the Ni formed a special electronic and coordination structure completely different from the metallic and oxide standards. The further EXAFS fitting of the steady state XAFS spectra was performed to reveal the detailed coordination structure of supported Ni species. The much stronger intensity

of the Ni–Ni coordination peak of Ni-4nm/CeO₂ catalyst in R-space FT-EXAFS spectra than that of the Ni-4nm/Mo₂N catalyst were observed (Fig. 4b), indicating the size of Ni species on γ-Mo₂N is much smaller than that on the CeO₂ substrate. Indeed, based on the EXAFS fitting results (Supplementary Table 1 and Supplementary Figs. 4–5), the C.N.(Ni–Ni) of Ni-4nm/CeO₂-590 was 10.8. While the C.N.(Ni–Ni) of the Ni-4nm/ γ-Mo₂N-590 was only 4. To track the temperature effect on the coordination shell of the Ni domains on the molybdenum nitride supports, the EXAFS spectra of Ni-4nm/γ-Mo₂N-400 at the different temperatures were also collected and presented in Fig. 4c. It suggested that the supported Ni was fully reduced after 400 °C reduction, as the major neighbor atoms of Ni atoms located at ~2.49 Å, corresponding to a typical metallic

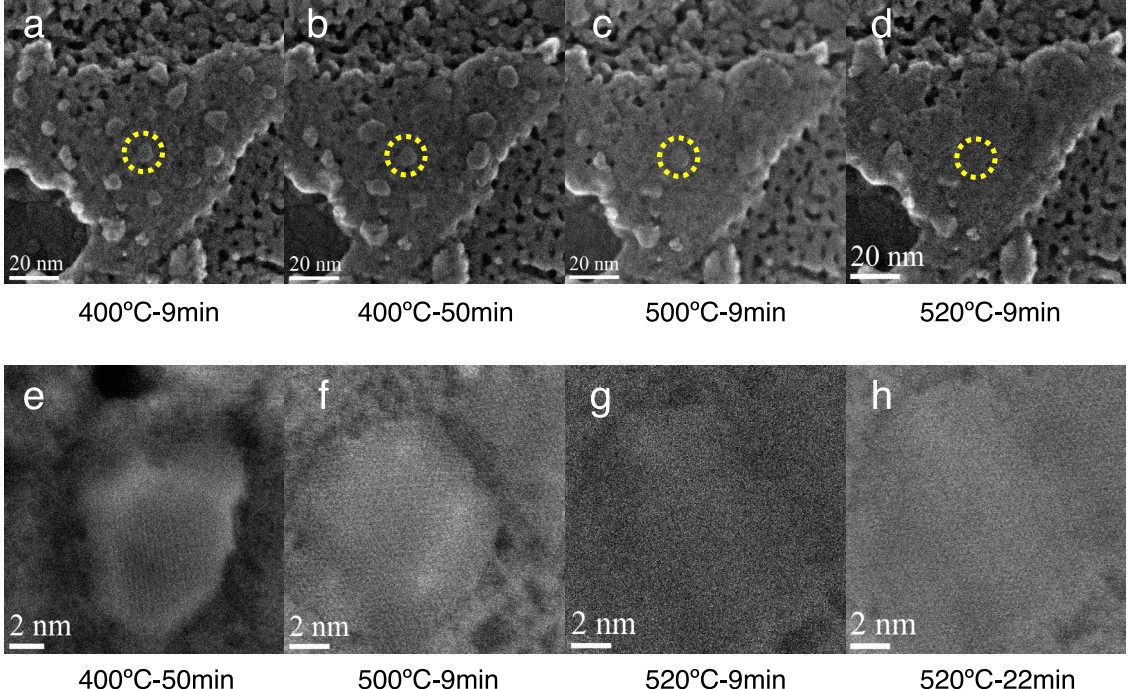

**Fig. 5 Environmental SE/STEM characterization.** The evolution of Ni particles on the support of γ-Mo₂N in the flow of H₂/N₂ at a ratio of 3:1 under different pretreatment temperature, **a–d** large scale SE/STEM images of the Ni-4nm/γ-Mo₂N catalyst at 400 °C for 9 min, 400 °C for 50 min, 500 °C for 9 min and 520 °C for 9 min, **e–g** high resolution images of selected region (marked as yellow circle in **b–d**, **h** high resolution images of Ni-4nm/γ-Mo₂N catalyst at 520 °C for 22 min.

Ni–Ni coordination. In addition, a peak appeared at 1.90 Å in the R space spectrum was identified as the Ni–N coordination (C.N.$_{Ni-N}$ of 1.1). This feature confirmed that the metallic Ni particles were located on the N-interlayer which was predicted by the AIMD simulations (Fig. 1a). With the elevated temperature, a novel Ni–Mo bonding at 2.62 Å appeared in the Ni/γ-Mo₂N-590 catalyst. These phenomena also indicated that the Ni has formed direct interaction with the molybdenum nitride support, confirming the similar structure variation as the AIMD simulations presented.

**Environmental SE/STEM studies of the structure evolution.** The spectroscopic evidence has given a detailed description on the electronic properties and coordination environment changes of the reverse sintering of supported 4 nm Ni particles over the molybdenum nitride surface. Electron microscopy is expected to directly monitor the structure evolution of the 4 nm Ni particles downsizing into the raft-like clusters. However, in conventional in-situ TEM/STEM experiments, thin and light supports, for instance, activated carbon, should be used or one has to find a proper orientation where electron beam is parallel to interface between nanoparticles and supports[35,36]. In this work, the relatively lower Z-contrast of Ni particles than the Mo makes it difficult for the conventional environmental scanning transmission electron microscope (STEM) imaging techniques to distinguish the Ni from the γ-Mo₂N substrate[28]. Moreover, it is hard to derive the surface information from the recorded 2D projection when samples were thick (above 100 nm) or in irregular shape. The new electron microscopy technique of environmental probe-corrected scanning transmission electron microscope equipped with secondary electron detector was applied to achieve the simultaneous acquisition of SE image and STEM image (STEM-ADF, STEM-BF) with an atomic spatial resolution (below 1 Å)[37]. Utilizing the surface sensitive low energy secondary electron caused by interaction between the primary beam and object, SE images showed a powerful ability to analyze surface morphology on bulk materials, regardless of the thickness and Z contrast of the metal on the support[38]. With the assistance of the SE-STEM method, we managed to observe the structural evolution of Ni particles on γ-Mo₂N directly in-situ during the high temperature treatment under the designated atmosphere.

As shown in Fig. 5a, the γ-Mo₂N support is highly porous, and the supported nanoparticles were well dispersed on the surface of the γ-Mo₂N at 400 °C in the simulated activation atmosphere. The corresponding BF and ADF-STEM did not show useful information since the support is too thick (Supplementary Fig. 6). On the SE images, the supported nanoparticles showed an irregular polygon shape (~5–7 nm) with clear edge (Fig. 5). No significant morphology change was observed with the time evolution at 400 °C (Fig. 5a, b). The detailed analysis of the fast Fourier transformation (FFT) of SE and BF images of the Ni-4nm/γ-Mo₂N at 400 °C were shown in Fig. 6a, b. Due to the penetration depth differences of secondary electron and traditional electron probes, the FFT patterns of SE and STEM reflect the structural information of the near surface species and the bulk of the sample respectively. As shown in the Fig. 6a, the region 1 shows a typical pattern of FCC structured γ-Mo₂N with a d-spacing ~2.0 Å, which is the bare support. While region 2 is confirmed unexpectedly as Ni₄N particles (d-spacing 2.6 Å, Supplementary Fig. 7) rather than the metallic Ni. This phenomenon suggested that the formation of Ni–N bonding changes the bulk structure of loaded Ni species. While the FFT of the BF image at region I is identical to the SE one, confirming the bulk phase of region 1 is the substrate. In contrast the features of both γ-Mo₂N and Ni₄N could be seen at region II in the FFT of STEM images, demonstrating the bulk of the catalyst below region II is composed by both the Ni₄N particles and the nitride support. These results also demonstrate that the loaded Ni particles are 3D particles with considerable thickness at 400 °C.

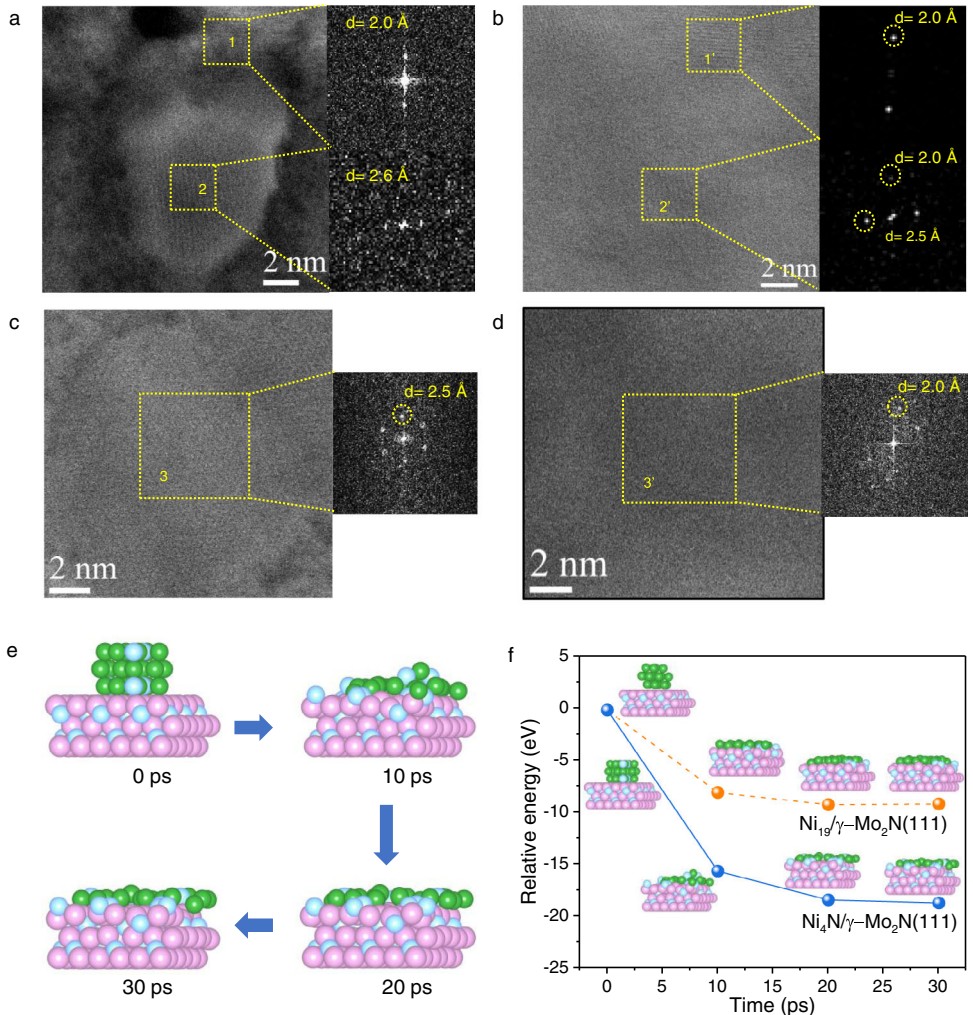

**Fig. 6 Comparison of high-resolution environmental SE and BF images.** Corresponding FFT patterns of high-resolution (**a**, **c**) SE and (**b**, **d**) BF image from the selected area at (**a**, **b**) 400 °C and (**c**, **d**) 520 °C. Area 1, 2, and 3 marked in the SE images and the Area 1', 2', and 3' marked in the BF images are the at the same position of the catalysts. **e** The structure evolution of $Ni_4N$ ($Ni_{20}N_5$ cluster) on $\gamma$-$Mo_2N$(111) in AIMD calculation in 30 ps; and **f** the interaction of the free energy change of the redispersion process of $Ni_{19}$/$\gamma$-$Mo_2N$(111) and $Ni_4N$/$\gamma$-$Mo_2N$(111). In this figure, the green, pink, and light blue atoms are Ni, Mo, and N respectively.

With the elevated activation temperature, the supported nanoparticles on the surface begins to fade, or even disappear. The SE images of 520 °C (Fig. 5d, g, h) proves that the pores in the $\gamma$-$Mo_2N$ substrate have disappeared, probably been covered by the loaded species, indicating the Ni has formed raft-like clusters and spread over the support (Supplementary Movie 9). The FFT patterns of SE and STEM (Fig. 6c, d) further confirmed the thermal induced redispersion phenomenon. The SE FFT pattern at region III showed a typical $Ni_4N$ features with d-spacing ~2.5 Å, suggesting the surface of region 3 is still covered by the Ni species. However, the STEM FFT pattern at the same region only showed features belong to $\gamma$-$Mo_2N$, which demonstrates that the Ni particle is possibly too thin to generate diffraction patterns. Compared with the electron diffraction patterns collected at 400 °C, we could confirm that the Ni particles have been spread into 2D layers on the nitride substrate under the thermal treatment. The normalized intensity profiles of the 520 °C sample's surface were much weaker than those of the 400 and 500 °C samples (Supplementary Fig. 8), which is another evidence that the Ni particles have reverse sintering at high temperature. The EDS mapping on STEM of ex-situ samples of Ni-4nm/$\gamma$-$Mo_2N$-590 catalysts were also carried out to further confirm the

spontaneous dispersion of Ni NPs on the $\gamma$-$Mo_2N$ (Supplementary Fig. 9).

As the SE-STEM characterization has confirmed the existence of $Ni_4N$ during the thermal treatment, it is important to further confirm whether the $Ni_4N$ is able to further reverse sintering to undercoordinated structures. The AIMD simulation has been done using the same setting to the $Ni_{19}$/$\gamma$-$Mo_2N$ over a novel $Ni_4N$ ($Ni_{20}N_5$ cluster was truncated from bulk $Ni_4N$($Pm\bar{3}m$)/$\gamma$-$Mo_2N$ model surface (Fig. 6e and Supplementary Movie 10–11). The spherical $Ni_4N$ turns directly into layered structure with maximized Ni–N-Mo bonding with in 10 ps time evolution (Supplementary Fig. 10). The relative stabilization energy is calculated as 18 eV, even larger than the energy change of $Ni_{19}$/$\gamma$-$Mo_2N$ system (Fig. 6f), ensuring that the loaded Ni particles will be converted into layered structure under thermal perturbation.

**Catalytic performance of Ni/$\gamma$-$Mo_2N$ in $CO_2$ hydrogenation.** The special chemical properties of the under-coordinated Ni species derived from the strong interaction between supported Ni and the molybdenum nitride substrates have been utilized in the catalytic hydrogenation of carbon dioxide. It is a common sense

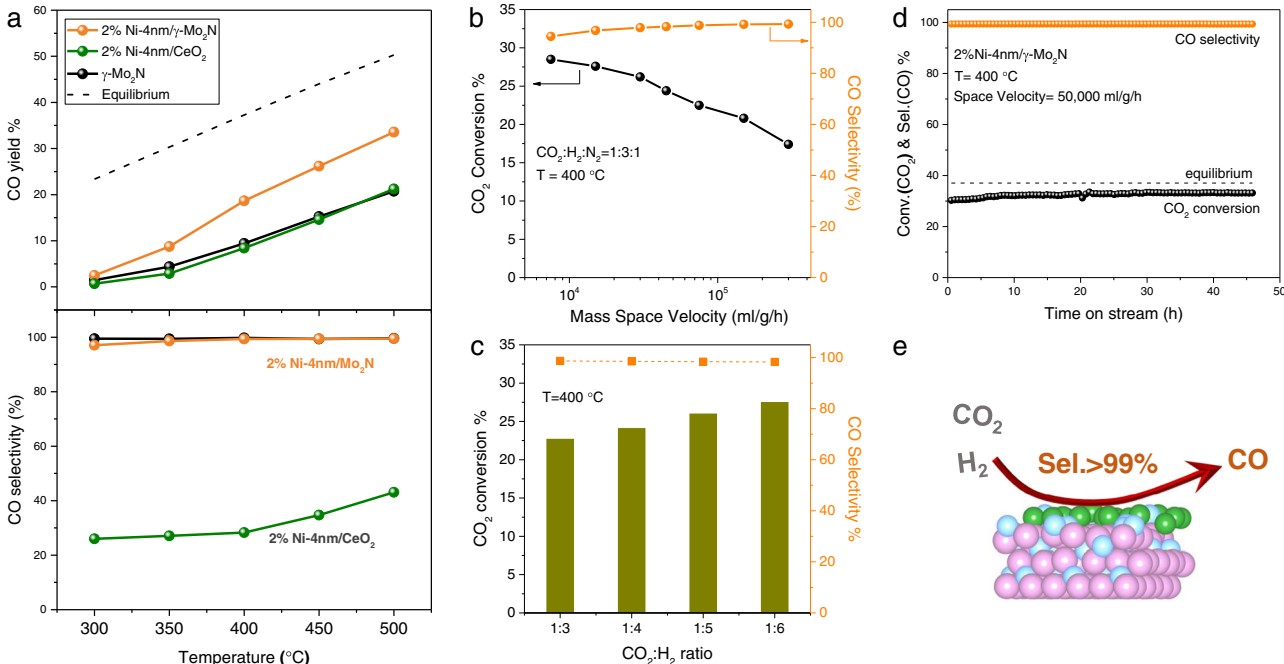

**Fig. 7 Effect of reaction conditions on CO$_2$ hydrogenation over the Ni-4nm/γ-Mo$_2$N catalysts. a** Effect of temperature on CO$_2$ hydrogenation over Ni-4nm/γ-Mo$_2$N, Ni-4nm/CeO$_2$, and γ-Mo$_2$N catalysts (CO$_2$:H$_2$:N$_2$ = 1:3:1, GHSV = 300,000 mL g$_{cat}^{-1}$ h$^{-1}$); Effect of **b** varied mass space velocity at a ratio of CO$_2$:H$_2$:N$_2$ at 1:3:1 and **c** varied CO$_2$/H$_2$ ratio at reaction temperature of 400 °C; **d** the stabilities evaluation of Ni-4nm/γ-Mo$_2$N catalysts in the CO$_2$ hydrogenation (T = 400 °C, CO$_2$:H$_2$:N$_2$ = 1:3:1, GHSV = 50,000 mL g$_{cat}^{-1}$ h$^{-1}$); **e** The schematic illustration of CO$_2$ hydrogenation over layered Ni-4nm/γ-Mo$_2$N. In this figure, the green, pink, and light blue atoms are Ni, Mo, and N respectively.

that the Ni-based catalysts were highly active in the methanation reactions of CO and CO$_2$ with exclusive high selectivity to the product methane[39–42]. Ever since the discovery of the CO$_2$ methanation by Sabatier and coworkers in 1902[43], Ni has been considered to be the typical inexpensive metal for this hydrogenation conversion process. Mechanistic studies have confirmed that the domains of metallic Ni particles should be above 2 nm in order to achieve the optimal methane formation rate and efficiency[40,42]. Single site Ni$_1$ and sub-nm Ni$_n$ sites are necessary to reverse the selectivity of Ni-based catalyst from CH$_4$ to CO in CO$_2$ hydrogenation. However, when the reaction temperature increased from 300 to 350 °C, the single site of Ni$_1$/MgO was starting to aggregate to particles with the selectivity of CH$_4$ appeared to 40%[44]. Indeed, the production of methane has been significantly suppressed on the activated Ni-4nm/γ-Mo$_2$N catalyst with highly dispersed Ni species. In the range from 250 to 500 °C, the selectivity of CH$_4$ has never exceeded 3%, even at the low temperature region at which the reverse water-gas shift reaction is thermodynamically unfavorable. When the working temperature went above 400 °C, the CO$_2$ was chemoselectively converted into CO (S$_{(CO)}$ > 99%) (Fig. 7a). While in comparison, when using the activated Ni-4nm/CeO$_2$ as the catalyst, the selectivity of methane was over 70% at 400 °C (CO$_2$ conv. ~8%), and ~60% at 500 °C (CO$_2$ conv. ~12%). What's more, the extraordinary selectivity toward CO remained above 95%, when decreasing the mass space velocity to 7500 ml/g/h (Fig. 7b) or tuning the CO$_2$/H$_2$ ratio (1:3 to 1:6) in the gas feed (Fig. 7c). Under a typical condition, the activated Ni-4nm/γ-Mo$_2$N catalyst exhibited stable activity in the hydrogenation of CO$_2$ for over 50 h. Neither the CO$_2$ conversion decay nor the decreasing on the selectivity of CO has been observed (Fig. 7d). The completely selectivity changes of the Ni-4nm/γ-Mo$_2$N to Ni-4nm/CeO$_2$ catalysts suggested that the reverse sintering phenomenon induced by the strong interaction between the Ni and Mo$_2$N was

able to significantly change the intrinsic catalytic properties of Ni-based catalysts and prolong the stability of Ni catalysts (Fig. 6e and Supplementary Fig. 11). The reduced dimension and electronic deficiency of Ni species are the main reasons for the reduced Ni-4nm/Mo$_2$N catalysts active for the RWGS reaction other than methanation. It has also been demonstrated the reverse sintering effect in the Ni/γ-Mo$_2$N catalyst is able to enhance the water splitting activity by increasing the exposure of the active metal sites (Supplementary Fig. 12). Comparing the fresh catalyst with the Ni-4nm/γ-Mo$_2$N-590 catalyst, the hydrogen evolution rate increased by ~10 times. Therefore, the reverse sintering effect can be used to maximize the dispersion and utilization efficiency of metals using simple treatments.

In conclusion, we have demonstrated using theoretical calculation and ab initio molecular dynamic simulation methods that, the pre-synthesized 4 nm Ni particles are able to reverse sintering into under-coordinated Ni species after the high temperature activation procedure driven by the strong interaction between the γ-Mo$_2$N and the Ni. The existence of both reduced Ni particles and the bare γ-Mo$_2$N is important for the formation of the highly dispersed Ni species. In-situ structural characterizations have confirmed the dispersion of Ni occurred after the reduction of bulk phase Ni and the removal of surface passivated O-layer of γ-Mo$_2$N. The reverse sintering effect for Ni-4nm/γ-Mo$_2$N catalyst has a positive effect on the chemoselective hydrogenation of CO$_2$ to CO. Compared with the Ni-4nm/CeO$_2$ reference catalyst (CO Sel.%~29%), the activated Ni-4nm/γ-Mo$_2$N catalyst exhibited over 96% CO selectivity at an even higher CO$_2$ conversion. This reverse transformation in the catalytic performance compared with traditional Ni-based catalysts can accounts with the dispersion and wetting phenomena of Ni nanoparticle on γ-Mo$_2$N, which serves as an excellent example of the potential application of reverse sintering effect in the high temperature favorable reactions.

## Methods

**DFT calculation.** Ab initio molecular dynamic (AIMD) simulations were carried out with Vienna Ab intio Simulation Package (VASP)[45,46] for 30 ps at a time step of 1 fs. The canonical ensemble (NVT) and Nosé−Hoover thermostats were set to 590 °C[47,48]. The potential energies change suggested that all simulations reached equilibrium at last (Supplementary Fig. 1). The spin-polarized static DFT calculations were performed to get the relative energies of systems at different period among the AIMD simulations. The initial structures for static DFT calculations were from AIMD simulations and be converged to local minimization. In all simulations, the electron exchange and correlation energy was treated within the Perdew-Burke-Ernzerhof (PBE)[49] functional of generalized gradient approximations (GGA)[50]. The election–ion interaction was described by the projector augmented wave (PAW) method[51], and the iterative solutions of Kohn-Sham equations was done using a plane-wave basis set with a cutoff energy of 400 eV. The sampling of the Brillouin zone was performed using a Monkhorst-Pack scheme with Gamma only K-points. In static calculations, the electronic relaxation criterion is that the change in total energy between two successive steps should be <$1.0 \times 10^{-4}$ eV per atom and the forces were <0.05 eV/Å.

The interface area was calculated according to equation R1,

$$S_{interface} \approx \frac{N'_M}{N_M} \times S \qquad (1)$$

where $N'_M$ is the number of surface metal atoms (Mo or Ce) which connected to Ni, $N_M$ is the total number of surface metal atoms of support, and $S$ is the surface area. The relative interface area is the ratio in function to time is plotted, the initial interface area was taken as a benchmark as "1".

In calculations, the bulk structure of $Mo_2N$ is in accordance with previous study[28]. For slab model, the Mo-terminated $Mo_2N(111)$ surface with ($6 \times 6$) supercell was used, there are 108 Mo, 36 N and 19 Ni atoms in $Ni_{19}/Mo_2N(111)$ model and 108 Mo, 36 N, 19 Ni, and 31 O atoms in $Ni_{19}O_{19}/O-Mo_2N(111)$ model. In $Ni_4N/Mo_2N(111)$ model, supported $Ni_{20}N_5$ cluster was truncated from bulk $Ni_4N$ (Pm-3m), so there are 108 Mo, 41 N, and 20 Ni atoms. In these three models, 36 Mo and 18 N atoms were fixed, while other atoms were relaxed. For $CeO_2$ support, an O-terminated $CeO_2(111)$ surface with ($5 \times 5$) supercell was used, it contains 50 Ce, 100 O and 19 Ni atoms, where 25 Ce and 25 O atoms were fixed and others were relaxed.

### Catalyst preparation

*Material.* Nickel acetylacetonate (Ni(acac)₂, 95%), oleylamine (OAm, 70%), tributylphosphine (TBP, 97%), and ammonia molybdate (($NH_4$)₆$Mo_7O_{24}$·4$H_2O$) were purchased from Sigma-Aldrich and used without any further purification. Benzyl ether (BE, 99%) was bought from Acros Organics and used without any further purification. Isopropanol and hexane were purchased from Fisher (ACS Certified), distilled before use and kept with molecular sieve. All experiment was conducted under standard Schlenk line condition.

*Synthesis of Ni-4nm[33].* Ni(acac)₂ (600 mg) was mixed with BE (60 ml) and OAm (8 ml) firstly. The mixture was stirred under vacuum for 5 min and the reaction was switched to $N_2$ atmosphere then. In all, 5 ml TBP was injected to the reaction flask. The mixture was further heated under vacuum at 100 °C for 1 h, generating a dark green transparent solution. The reaction flask was then filled with a $N_2$ blanket and the green solution was heated to 230 °C at a heating ramp of 5 °C/min. The reaction was kept for 15 min at this temperature. All the product was transferred to $N_2$-filled glovebox to avoid exposure to air. The product was purified with extra dry isopropanol and separated by centrifugation (9016×*g*, 8 min). The purification was repeated two times and the product was then dispersed in extra dry hexane for further application.

*Preparation of γ-$Mo_2N$.* First, $MoO_3$ powders were synthesized by calcination of ($NH_4$)₆$Mo_7O_{24}$·4$H_2O$ at 773 K for 4 h. Then, the $MoO_3$ powder was grounded and the fine powder of $MoO_3$ was transferred to a quartz tube and ammonized under the flow of pure $NH_3$ (150 mL/min) at 973 K for 4 h (heating rate was set as 5 K/min). After cooling down to room temperature, the obtained γ-$Mo_2N$ catalyst was passivated with $CO_2$ at room temperature overnight to protect the as-synthesized γ-$Mo_2N$ material from the pyrophoric oxidation when contacted with air.

*The protocol of Ni/γ-$Mo_2N$ synthesis.* The Ni-4nm/γ-$Mo_2N$ catalysts were synthesized by using incipient wet impregnation (IWI) of as-prepared homogeneous Ni-4nm hexane solution over the as-synthesized γ-$Mo_2N$. The slurry was then freeze-dried under stirring. The loading of Ni loading was set to ~2%. The catalyst is denoted as Ni-4nm/γ-$Mo_2N$.

The dispersion phenomena and mechanism of Ni-4nm/γ-$Mo_2N$ was investigated in the flow of the $N_2$-$H_2$ mixture (1:3 v/v, 32 mL/min) at a temperature ramp of 5 °C/min to 590 °C.

### Catalysts characterization

*QXANES and XAFS.* The in Situ Quick X-ray Absorption Near Edge Spectroscopy (QXANES) spectra was collected at ISS beamline of the Brookhaven National Laboratory (BNL), and X-ray Absorption Fine Structure Spectroscopy (XAFS) of

the catalysts was collected at the 9-BM beamline of the Advance Photon Source (APS), Argonne National Laboratory (ANL). In the test, the powder samples were loaded in a Clausen Cell. The Ni K-edge XAFS spectra were collected in the fluorescence mode using a PIPS detector at BNL and four-channel vortex detector at APS, respectively. The activation processes for QXANES experiments is carried out in the flow of $N_2$/$H_2$ mixture (v:v = 1:3) at a temperature ramp of 2 °C/min to 590 °C. Each scan takes 1 min, and every 10 scans were merged together to get a spectrum. For the XAFS experiments, The activation processes were performed in the flow of $N_2$/$H_2$ mixture (v:v = 1:3) at a temperature ramp of 10 °C/min, but at the temperature of 400 and 590 °C is held for 40 min before collecting spectra. Each scan will take 20 min, and every three scans are merged to be one spectrum. The data pretreatment and extended X-ray absorption fine structure (EXAFS) fitting were performed using the Ifeffit package[52].

*Ambient pressure X-ray photoelectron spectroscopy.* AP-XPS spectra were collected by a SPECS AP-XPS chamber equipped with a PHOIBOS 150 EP MCD-9 analyzer with an energy resolution of 0.4 eV. The source of AP-XPS is the Mg Kα radiation (1253.5 eV). The C 1s photoemission line with the surface carbon feature (284.8 eV) was used for the binding energy calibration. The 2%Ni-4nm/γ-$Mo_2N$ catalyst were pressed into a clean aluminum plate as XPS samples. Typically, the fresh sample was activated in the AP-XPS chamber by a 40 mTorr $N_2$/$H_2$ (1:3 v/v) mixture at 300, 350, 400, 500, and 520 °C (the upper limit of the instrument) for 1 h. The spectra at the Ni 2p, Mo 3d/3p, O 1s, N 1s, and C 1s XPS regions were collected at each temperature after an hour treatment.

*Electron microscopy.* TEM, HAADF-STEM images and EDS mappings in Fig. 2 and Supplementary Fig. 4 were obtained using a FEI Talos F200X S/TEM microscope with a field-emission gun at 200 kV at CFN of Brookhaven National Lab.

*Environmental SE-STEM.* Environmental secondary electron/scanning transmission electron microscopy (Environmental SE/STEM) experiments were carried out using a probe aberration corrected Hitachi HF5000 equipped with a secondary detector, operated at 200 kV. The facility allows spontaneously record bright-field (BF), annual dark field (ADF), and secondary electron (SE) images under the STEM mode. By using the probe aberration corrector, it can get a resolution of 0.7 Å in ADF-STEM images and a resolution of below 1 Å in SE-STEM images. The sample was directly dispersed on the MEMS chip without any pretreatment. After inserting the heating holder equipped with the MEMS chip, a mixture of $H_2$ and $N_2$ with a molar ratio equal to 3:1 was introduced into the sample chamber area and a total pressure of 2 Pa was maintained during the whole experiments. The sample was directly heated to 200 °C prior to the in-situ study. Once the sample was heated to the desired temperature, the following changes in morphology and structure of the samples were imaged.

*Performance evaluation of $CO_2$ hydrogenation.* In a typical experiment, 10 mg of Ni-4nm/γ-$Mo_2N$ catalyst powder was mixed with 10 mg of pre-calcined $SiO_2$ and loaded in a fixed-bed reactor. The sample was activated in a flow of a $N_2$ and $H_2$ mixture (1:3 v/v) at 590 °C for 2 h. After activation, the temperature of the reactor was changed to a designated temperature in the $N_2$-$H_2$ mixture and then switched to the reaction gas feed ($CO_2$/$H_2$/$N_2$ 1:3:1 v/v/v). The same procedures were performed for the reference catalysts. The products of the reaction were analyzed by online gas chromatography (GC, Agilent 7890) equipped with a thermal conductive detector (TCD) and a flammable ionization detector (FID). The $N_2$ in the flow was used as the inner standard. The response factor of each reactant and product was calibrated using standard curve methods.

## Data availability

The data that support the plots within this paper and other finding of this study are available from the corresponding author upon reasonable request.

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

## Acknowledgements

This work was financially supported by the Natural Science Foundation of China (21725301 (D.M.), 21932002 (D.M.), 21821004 (D.M.), 21991153 (D.M.), 21872163 (X.L.), 22072090 (X.L.), and 22002140 (L.L.)) and the National Key R&D Program of China (2017YFB0602200 (D.M.) and 2021YFC2101800 (S.Y.)), and Zhejiang Provincial Natural Science Foundation of China under Grant (LR21B030001 (S.Y.)). The experiments of AP-XPS carried out at the Chemistry Department of Brookhaven National Laboratory (BNL) were supported by the division of Chemical Science, Geoscience, and Bioscience, Office of Basic Energy Science of the U.S. Department of Energy (DOE) under contract No. DE-SC0012704. Use of the Advanced Photon Source (beamlines 9-BM, for XAS characterization) was supported by the U.S. DOE under contract no. DE-AC02-06CH11357. Young Elite Scientist Sponsorship Program by CAST, NO. 2019QNRC001 is also acknowledged by L.L. We also appreciate technical supports from Mr. Hiroaki Matsumoto and Mr. Chaobin Zeng, Hitachi High-Technologies (Shanghai) Co. Ltd, for HR-STEM characterization.

## Author contributions

D.M., J.R., and L.L. designed the study. L.L. performed most of the experiments and data analysis. Y.W. carried out the experiments of catalytic performance evaluation. X.L. and F.Y. carried out the measurement and analysis of environmental SE/STEM images. J.L. and X.W. did the DFT calculation. Z.G., N.R., M.W., and S.S. did the AP-XPS measurement. S.Y. and F.Z. did the in-situ XAFS measurement at APS. C.L. and S.Z. did the synthesis of uniform Ni-4nm particles. L.H. carried out the measurement of TEM and STEM. L.L., S.Y., J.R., and D.M. wrote the paper. X.L. discussed and revised the paper.

## Competing interests

The authors declare no competing interests.
