## [Peer Review File · Nature Communications]

Title: Reversing sintering effect of Ni particles on γ -Mo₂N via strong metal support interactionREVIEWER COMMENTS

Reviewer #1 (Remarks to the Author):

The paper of Ma et al. describe the reversing sintering transformation from metal nanoparticles to clusters, which is highly interesting for catalytic applications, allowing to obtain activities in the order of small particle sizes. This behaviour has been exemplified using 4nm Ni NP loaded on γ -Mo₂N. Under reductive treatment, it transforms into rafted like Ni clusters. According to the authors, the re-dispersion of the metal nanoparticle requires that the metal-support interaction exceed the metal-metal interaction inside the nanoparticle. The results presented in this work are in line with previous studies where a reversible transformation between single atoms/clusters and nanoparticles have been found under reduction-oxidation treatments. The manuscript is interesting and the result of great interest for the scientific community. I support its publication in Nature Communications. However, I have some comments which should be addressed by the authors before publication.

The re-dispersion process was followed by in situ X-ray absorption spectroscopy (XAS), X-ray photoelectron spectroscopy (XPS), environmental scanning transmission electron microscopy/secondary electron (ESTEM/ESE) and DFT calculation. In fact, the formation of Ni-N bonds are observed by in situ EXAFS studies after hydrogen treatment at 400 °C, while at higher temperature, i.e. 590 °C Ni-Mo start to be formed. These results support the formation of raft like Ni species. However, this behaviour is difficult to see by AP-XPS. On one hand, the authors should analyse the N1s core line, and try to find out the formation of Ni-N species, as well as display the variation in the Ni:N:Mo surface composition under different treatment conditions. Moreover, I would expect changes in the Mo3d and Ni2p lines supporting the EXAFS data. In addition, the specification Ni₄N as reflected in the caption of figure 6, confuse the reader, because EXAFS data gives an average coordination number Ni-N 1.7 (table S1). Another question is regarding beam damage in the TEM studies. Can the loss of signal detected in Figure 5(d and h), be ascribed to beam damage?.

The authors use AP-XPS, please indicate the energy of the X-ray radiation.

Regarding the catalytic data, I would like to see the activity of the support. Mo₂N is reported to be active in the CO₂ hydrogenation, and it needs to be included. In addition it would be interesting to characterize the catalyst after reaction (spent samples) or if possible under reaction. It is true that the catalyst displays good stability with TOS; but, since water is formed under reaction conditions, it would be interesting to analyse the resistance of the active site in the presence of water and high temperature, and/or to perform stability tests at lower space velocity (i.e higher conversion degrees).

Finally, is the re-dispersion reversible? And if yes, under which conditions? This information will allow the readers to define the applicability of the method to specific catalytic applications.

Reviewer #2 (Remarks to the Author):

Ma et al. have prepared a very interesting and thorough manuscript on Ni nanoparticles supported on γ -Mo₂N. The ~4 nm nanoparticles restructure after high temperature treatment in H₂ to form a highly

active catalyst for the reverse water-gas shift reaction. The catalysts are characterized using state-of-the-art in situ techniques and demonstrate significant improvement versus the Ni/CeO₂ control. Although the characterization of the catalysts are quite thorough, there are some minor points that need to be addressed before publication can be recommended in Nature Communications.

1. The time scale of the restructuring to form the Ni nano-rafts in the AIMD calculations seems very fast relative to what is observed in the environmental TEM images in Fig. 6. This suggests that the authors may have made an assumption in their theoretical model that does not necessarily represent the actual electronic interactions between the Ni and Mo₂N. Are there other Ni structures that could be present in the Ni/Mo₂N catalyst besides the nanorrafts? The formation of highly dispersed Ni within the Mo₂N lattice could be difficult to distinguish between the Ni rafts in the XAFS and TEM. It would be worthwhile for the authors to consider alternative Ni structures in their discussion because the experimental evidence for the formation of Ni rafts is not fully convincing.

2. The Ni/Mo₂N catalysts are highly active for RWGS when Ni catalysts are typically active for the Sabatier reaction. Do the authors have mechanistic insight as to why these catalyst are more active for RWGS even at slower GHSV and higher H₂:CO₂ ratios? Additional insight into the mechanism would greatly strengthen the manuscript.

3. The reactor data in Figure 7 is a little confusing as all of the conditions are not listed in the figure caption. The baseline conditions of GHSV, CO₂:H₂ ratio and CO₂ conversion (7a) are not always shown for each sub-figure. Including the full details of the reaction conditions would help readers better evaluate the catalytic performance.

4. Including the Ni/CeO₂ catalyst in the manuscript is useful to benchmark the performance against a standard catalyst. However, because of the high performance of the Ni/Mo₂N catalyst for RWGS, it would also be important to include details of other single atom Ni catalysts for comparison. For example, Frei et al 10.1021/jacs.8b11729 exhibits similar performance to the current work. Comparison to other Ni single atom catalysts as a benchmark is necessary in addition to the Ni/CeO₂ control.

5. The language in the manuscript requires some revision due to grammatical errors and is a bit too colloquial for a journal article. At times this hinders understanding of the manuscript. For example, 'exhibits a remarkable catalytic selectivity reverse compared with traditional Ni based catalyst...' (abstract), 'a strong capture ability to grasp' (P2), 'tough reaction conditions' (P3), 'spit metals from 3D nanoparticles' (P4).

Reviewer #3 (Remarks to the Author):

The manuscript details a joint experimental-theoretical study into the formation of layer-like Ni-based metal nanoparticles on a molybdenum nitride substrate, via high temperature reduction of supported (oxidized) Ni particles. STEM, coupled with Ab Initio simulations suggest that reduced Ni (and Ni₄N) particles spontaneously form island-like layers, due to a strong metal support interaction (SMSI) effect, which drives a change in catalytic selectivity towards CO₂->CO conversion. The findings are reasonable and supported by a range of complementary techniques. There is however, little discussion of the significance or generality of the findings, or their place in the context of other work, which is rather under-cited generally. For example, under-coordinated Nickel has been shown in several works to increase selectivity towards CO₂->CO conversion (e.g. [10.1039/C8EE00133B](https://doi.org/10.1039/C8EE00133B), [10.1021/acs.accounts.7b00634](https://doi.org/10.1021/acs.accounts.7b00634)). The computational setup is rather incompletely described, and so it is difficult to make a judgement on the quality of the findings given. There are several areas in which I believe the manuscript would benefit from clarification or additional analyses. These are listed below.

1) No justification is presented for using Ni₁₉ as the analogue of a 4 nm Ni particle. The existence of quantum mechanical finite size effects in sub-nm/1 nm metal particles are well known, and often differ significantly from several nm particles. Hence Ni₁₉ is not *a priori* a reasonable model of the systems used in experiment. Some computational tests of a larger particle (even with static calculations) would be very useful to make this case.

2) The timeframe of the simulations is very short. It appears in the case of Ni@Ceria that the energy and the structure are not converged, and the energy trends for NiO@O-Mo₂N are not shown on the plot. Hence conclusions about the propensity towards surface wetting and the energy of the process are currently questionable. Longer simulations are needed to really make the claims in the manuscript. It would also be of value to report an estimate of the interface surface area as a function of time.

3) Some tests are required to verify that the small depth of the slabs, the fixed portion in the lower layer, and the lateral dimensions of the slabs are sufficient. In the case of Ni₄N, for example, the formed islands clearly interact across the supercells, which may affect the relative energies. It is therefore necessary to determine, even indirectly via static calculations, the extent of this effect.

4) The charge analyses are currently unclear. No units are given for charge depletion/accumulation on figure 3. In addition, charge transfer from Ni to Mo₂N is invoked in the manuscript by inspection of the DFT charge density plots – but these densities only show an accumulation in the interlayer region. This is an expected outcome of Ni-surface bonding. It is not necessarily a reflection of electron deficiency in the particles, or accumulation in the substrate. To claim electron density is lost from the cluster and gained by the substrate, total charges projected onto the cluster or surface could be calculated via some standard charge partitioning scheme (such as the Bader method).

5) How are the relative energies calculated? NVT AIMD returns internal energies, which will fluctuate significantly about a mean. This mean is likely to vary a lot over the first few ps (and no pre-equilibration is mentioned in this manuscript). A plot of the total energy against time would be useful, along with a description of how the relative energy is extracted.

6) The title, and the main conclusion of the paper is that “reverse sintering” is achieved by reduction of Ni particles. But this is not strictly valid. What is shown is a 3d to 2d transition to form layers/islands. No evidence is presented that the Ni particles redisperse into small

clusters or single atoms. Perhaps a different choice of terminology would be more appropriate.

7) It is claimed that the “reverse sintering” increases catalyst longevity, but there is no comparison to base this upon.

8) I would appreciate some discussion/speculation of why the selectivity of the catalyst changes upon high temperature reduction. The oxidation state and the physical structure can both play roles, but this is not disentangled in the manuscript.

9) The referencing is a little lacking. There is a literature on the topic of undercoordinated metals supported on Mo_nX_m ($X=\text{N},\text{C}$) for catalysis, either deposited as layers or dispersed via reduction, e.g. 10.1021/jacs.8b08246 and references within, as well as on the topic of M@oxide SMSI effects. The concept of electronically-induced 3D to 2D transitions of supported metal particles has also been covered extensively, for example by Pacchioni, Freund and colleagues. I would ask the authors to provide a more thorough reference list and put the findings in this context.

10) Figure S5 is difficult to interpret. Is the SE intensity increasing between 400 and 500 C? Improved labelling of the figure would help.

A point-by-point response

We have addressed all the comments point-by-point and revised the manuscript accordingly. In this response letter, comments from referees are in black typeface, and our responses are started with **Reply** typeface. All major changes have been highlighted in blue in the main text.

Major changes in the manuscript: The AIMD simulations include in the manuscript and SI have been updated. The time scale has been prolonged to 30 ps to provide a more accurate prediction on the structure evolution. The energy profiles and interface areas have been provided to quantify the reverse sintering effect.

Reviewer #1 (Remarks to the Author):

“The manuscript is interesting and the discovery is of great interest for the scientific community. I support its publication in Nature Communications.”

1. These results of XAFS support the formation of raft like Ni species. However, this behaviour is difficult to see by AP-XPS. On one hand, the authors should analyse the N1s core line, and try to find out the formation of Ni-N species, as well as display the variation in the Ni:N:Mo surface composition under different treatment conditions.

Reply: Thanks for the review’s comments. Indeed, it is difficult to see the formation of raft like Ni species from AP-XPS, as XPS method is not a technique for geometric structure analysis. However, it could be observed from the Ni2p spectrum that the loaded Ni species exhibit electron deficient state ($\text{Ni}^{\delta+}$) with the increased treatment temperature (Fig. 3b), which is a strong evidence for the existence of SMSI.

The N 1s core line has been analyzed and adds as the Fig. 3b. It is hard to attribute the position of Ni-N species, mainly due to the strong overlap of the N 1s and Mo 3p signals (Fig. R1). As the Ni-N bonding takes only a small part of the total surface N, it is not very reliable to resolve the Ni-N species only.

Figure R1. The AP-XPS spectra of Ni-4nm/g-Mo₂N catalysts at Ni 2p, N1s/Mo3p and Mo 3d

regions.

2. In addition, the specification Ni₄N as reflected in the caption of figure 6, confuse the reader, because EXAFS data gives an average coordination number Ni-N 1.7 (table S1).

Reply: It should be noticed that the activation condition of the in-situ XAS and the SE-EM are quite different from each other. In addition, the XAFS is a technique which reflects the average structure of all Ni species, while EM can only analyze limited number of particles. Therefore, the inconsistency between two techniques can be explained by the heterogeneity of powder catalysts. The relatively large Ni₄N raft particles may contribute more to the Ni-N coordination number, while the highly dispersed Ni clusters has much smaller Ni-N coordination.

3. Another question is regarding beam damage in the TEM studies. Can the loss of signal detected in Figure 5(d and h), be ascribed to beam damage?

Reply: Thanks for the review's comment. We are sorry that we didn't give detailed information in the experiment part. We carefully monitored beam irradiation influences during the whole in-situ observation. The in-situ TEM experiments were mainly conducted under the 'dark' environment as we closed gun valve to reduce beam damage. We only opened gun valve to image the selected areas timely and closed gun valve immediately after the characterizations. During the imaging process (<10s), we didn't detect any changes in shape and microstructure. We revise the experimental part correspondingly.

4. The authors use AP-XPS, please indicate the energy of the X-ray radiation.

Reply: Thanks for the review's comment. The source of AP-XPS is the Mg K α radiation (1253.5 eV).

5. Regarding the catalytic data, I would like to see the activity of the support. Mo₂N is reported to be active in the CO₂ hydrogenation, and it needs to be included.

Reply: Thanks for the review's comment. The activity of Mo₂N support has been included in Figure 7a (black dot line) in the revised manuscript as shown in Figure R2.

Figure R2. The catalytic performances of 2%Ni/ γ -Mo₂N, 2%Ni/CeO₂ and γ -Mo₂N catalysts on CO₂ hydrogenation reaction.

6. In addition, it would be interesting to characterize the catalyst after reaction (spent samples) or if possible under reaction.

Reply: Thanks for the review's comment. The STEM-HAADFA and EDS mapping characterization of Ni/Mo₂N after long term stability test was carried out as shown in Figure R3, indicating the structure of dispersed Ni particles on Mo₂N was remained under reaction condition. It has been added as Fig. S11.

Figure R3. STEM-HAADF image and EDS mapping of Ni/ γ -Mo₂N after long term CO₂ hydrogenation reaction.

7. It is true that the catalyst displays good stability with TOS; but, since water is formed under reaction conditions, it would be interesting to analyse the resistance of the active site in the presence of water and high temperature, and/or to perform stability tests at lower space velocity (i.e. higher conversion degrees).

Reply: Thanks for the review's comment. As shown in Figure R4, the 2%Ni/Mo₂N shows good

stability at lower space velocity. When the WHSV reduces from 600,000 to 50,000 ml/g/h, the initial CO₂ conversion is 30.3%. After 44 hrs, the CO₂ conversion further increases to 33.1%. This result indicates that the highly dispersed Ni active sites have high resistance to the water in the gas phase.

Figure R4. Long term stability of Ni-4nm/ γ -Mo₂N on CO₂ hydrogenation at low space velocity of 50,000 ml/g/h.

8. Finally, is the re-dispersion reversible? And if yes, under which conditions? This information will allow the readers to define the applicability of the method to specific catalytic applications.

Reply: Thanks for the review's comment. To the best of our knowledge, the re-dispersion is reversible. But, the agglomeration for the dispersed Ni would occur in the oxidative atmosphere, as the interaction between NiO and oxygen modified nitride surface is weak based on our AIMD studies (Figure 1b). The re-dispersion will be happened under ammonia activation other than N₂/H₂ mixture treatment.

Reviewer #2 (Remarks to the Author):

Although the characterizations of the catalysts are quite thorough, there are some minor points that need to be addressed before publication can be recommended in Nature Communications.

1. The time scale of the restructuring to form the Ni nano-rafts in the AIMD calculations seems very fast relative to what is observed in the environmental TEM images in Fig. 6. This suggests that the authors may have made an assumption in their theoretical model that does not necessarily represent the actual electronic interactions between the Ni and Mo₂N. Are there other Ni structures that could be present in the Ni/Mo₂N catalyst besides the nanorrafts? The formation of highly dispersed Ni within the Mo₂N lattice could be difficult to distinguish between the Ni rafts in the XAFS and TEM. It would be worthwhile for the authors to consider alternative Ni structures in their discussion because the experimental evidence for the formation of Ni rafts is not fully convincing.

Reply: Thanks for the review's comments. Other structure like atomic dispersion has been considered, but it happens with less Ni atoms. In our calculation system, Ni₄ can be dispersed into atomic structure. The raft-like structure can be resolved from the experimental results of Ni/Mo₂N-4nm catalyst. From the XAFS characterization, the coordination number of Ni is changing from ~10.4 to 4.5. If Ni atoms doped into the lattice of the Mo₂N support, the C.N.(Ni-Ni) and C.N.(Ni-Mo) should be 0 and ~9 respectively. Meanwhile, SE images showed a powerful ability to analyze surface morphology on bulk materials, regardless of the thickness and Z contrast of the metal on the support. With the assistance of the SE-STEM method, the structural evolution of Ni particles on γ -

Mo₂N was directly observed during the high temperature treatment under the designated atmosphere. Based on our results, we think the structure described by the reviewer is not likely to occur in the Ni/Mo₂N-4nm system.

2. The Ni/Mo₂N catalysts are highly active for RWGS when Ni catalysts are typically active for the Sabatier reaction. Do the authors have mechanistic insight as to why these catalyst are more active for RWGS even at slower GHSVs and higher H₂:CO₂ ratios? Additional insight into the mechanism would greatly strengthen the manuscript.

Reply: Thanks for reviewer's suggestion. The Sabatier reaction has been widely demonstrated as a structure sensitive reaction. Reducing the continuity of the metallic particles is an effective method to inhibit the methanation selectivity in CO₂ hydrogenation reaction. The reduced dimension and positive electronic properties of Ni species are the main reasons for the reduced Ni-4nm/Mo₂N catalysts active for the RWGS reaction other than methanation. The brief mechanism discussion was added to the manuscript.

3. The reactor data in Figure 7 is a little confusing as all of the conditions are not listed in the figure caption. The baseline conditions of GHSV, CO₂:H₂ ratio and CO₂ conversion (7a) are not always shown for each sub-figure. Including the full details of the reaction conditions would help readers better evaluate the catalytic performance.

Reply: The experimental details of the catalytic performance evaluation have been revised according to the suggestions. The changes are highlighted in blue.

4. Including the Ni/CeO₂ catalyst in the manuscript is useful to benchmark the performance against a standard catalyst. However, because of the high performance of the Ni/Mo₂N catalyst for RWGS, it would also be important to include details of other single atom Ni catalysts for comparison. For example, Frei et al 10.1021/jacs.8b11729 exhibits similar performance to the current work. Comparison to other Ni single atom catalysts as a benchmark is necessary in addition to the Ni/CeO₂ control.

Reply: Thanks for reviewer's comment. The studies of Ni₁/MgO and other Ni based catalysts have demonstrated that the isolated Ni atoms are not able to convert CO₂ into methane, while Ni particles with sufficiently large diameters are the active sites of Sabatier reaction (cited as reference as 44). However, the Ni SACs supported over oxide supports are not stable at high temperature in reducing atmosphere. While our catalysts tend to stabilize the highly dispersed Ni species and enhance the dispersed Ni particles under the CO₂ hydrogenation reaction condition, which is a huge advantage over traditional metal/oxide catalysts.

5. The language in the manuscript requires some revision due to grammatical errors and is a bit too colloquial for a journal article. At times this hinders understanding of the manuscript. For example, 'exhibits a remarkable catalytic selectivity reverse compared with traditional Ni based catalyst...' (abstract), 'a strong capture ability to grasp' (P2), 'tough reaction conditions' (P3), 'spit metals from 3D nanoparticles' (P4).

Reply: Thanks for review's reminding. It has been corrected, and we revised the manuscript carefully to avoid the possible errors.

Reviewer #3 ((Remarks to the Author))

There is however, little discussion of the significance or generality of the findings, or their place in the context of other work, which is rather undercited generally. For example, under-coordinated Nickel has been shown in several works to increase selectivity towards CO₂->CO conversion (e.g. 10.1039/C8EE00133B, 10.1021/acs.accounts.7b00634).

The computational setup is rather incompletely described, and so it is difficult to make a judgement on the quality of the findings given.

Reply: The main novelty of our study is the reverse sintering effect of Ni particles on the Mo₂N under reductive reaction condition. RWGS reaction is the probe reaction to verify the structure changes. As you mentioned, that the under-coordinated Ni is able to increase the selectivity towards CO. Therefore, the reaction here is as an evidence to confirm our point that the Ni particles loaded with the initial diameter of 4 nm can reverse sintering on the Mo₂N to raft clusters under reductive condition. We also showed another catalytic probe reaction (electrochemistry reaction HER) which confirms the reverse sintering effect can increase the density of active site. It also highlights the strong interaction between Ni and Mo₂N support may be an efficient effect to influence the catalytic performance of loaded metal species.

1)No justification is presented for using Ni₁₉ as the analogue of a 4 nm Ni particle. The existence of quantum mechanical finite size effects in sub-nm/1 nm metal particles are well known, and often differ significantly from several nm particles. Hence Ni₁₉ is not a priori a reasonable model of the systems used in experiment. Some computational tests of a larger particle (even with static calculations) would be very useful to make this case.

Reply: Thanks for review's comments. We have simulated a Ni₅₅ particle supported on a 8*8 supercell of Mo₂N(111) slab for 30 ps (Video S7-8), the structure and potential energy change with time were presented in Figure R5. It could be found that the configuration change of Ni₅₅ on Mo₂N(111) is very similar to Ni₁₉ particle, the raft-like configuration is more favorable in thermodynamics.

Figure R5. (a) Potential energy of Ni₅₅/γ-Mo₂N(111) during AIMD simulation. (b) The relative energy change of Ni₅₅/γ-Mo₂N(111) from static DFT calculations.

2)The timeframe of the simulations is very short. It appears in the case of Ni@Ceria that the energy and the structure are not converged, and the energy trends for NiO@O-Mo₂N are not shown on the

plot. Hence conclusions about the propensity towards surface wetting and the energy of the process are currently questionable. Longer simulations are needed to really make the claims in the manuscript. It would also be of value to report an estimate of the interface surface area as a function of time.

Reply: Thanks for review's comments. We have prolonged the timeframe to 30 ps, the potential energy changes of each system with time were presented in Fig. R6 and added in Figure S1. The mean potential energies were slightly fluctuation at the final, indicating that the energy and the structure were converged after 30 ps simulation. As reviewer suggested, we also estimated the interface area according to equation R1, where N'_M is the number of surface metal atoms (Mo or Ce) which connected to Ni, N_M is the total number of surface metal atoms, and S is the surface area. The result was presented in Figure R7. For $Ni_{19}/\gamma\text{-Mo}_2\text{N}(111)$ and $Ni_4\text{N}/\gamma\text{-Mo}_2\text{N}(111)$, the interface surface area increases dramatically with time during the first 10 ps, and changes slightly with the time prolonged. But for $Ni_{19}/\text{CeO}_2(111)$ and $Ni_{19}\text{O}_{19}/\text{O-}\gamma\text{-Mo}_2\text{N}(111)$, the interface area increases a little after the AIMD simulations for 30 ps.

$$S_{\text{interface}} \approx \frac{N'_M}{N_M} \times S \quad (\text{Equation R1})$$

Figure R6. Potential energies of (a) $Ni_{19}/\gamma\text{-Mo}_2\text{N}(111)$, (b) $Ni_{19}\text{O}_{19}/\text{O-}\gamma\text{-Mo}_2\text{N}(111)$ and (c) $Ni_{19}/\text{CeO}_2(111)$ during AIMD simulations.

Figure R7. The interface area with function of time for (a) $Ni_{19}/\gamma\text{-Mo}_2\text{N}(111)$, (b) $Ni_{19}\text{O}_{19}/\gamma\text{-Mo}_2\text{N}(111)$ and (c) $Ni_{19}/\text{CeO}_2(111)$.

3) Some tests are required to verify that the small depth of the slabs, the fixed portion in the lower layer, and the lateral dimensions of the slabs are sufficient. In the case of $Ni_4\text{N}$, for example, the formed islands clearly interact across the supercells, which may affect the relative energies. It is therefore necessary to determine, even indirectly via static calculations, the extent of this effect.

Reply: Thanks for review's comment. We have carried out static DFT calculations and tested the depth of slab, fixed portion in the layer, and the supercells effect on cohesive energies (eV/atom)

between Ni₄N (Ni₂₀N₅) and γ -Mo₂N(111), the results were presented in Table R1. The results demonstrated that these factors have little effect on cohesive energies.

Table R1. The depth of slab, fixed portion in the layer, and the supercells effect on cohesive energies between Ni₄N (Ni₂₀N₅) and γ -Mo₂N(111).

	Supercells	Slab thickness (atomic layers)	Fixed layers (atomic layers)	Cohesive energy (eV/atom)
Entry 1#	6*6	5	2	-1.76
Entry 2#	6*6	5	0	-2.02
Entry 3#	8*8	5	2	-2.15
Entry 4#	6*6	7	2	-1.64

4) The charge analyses are currently unclear. No units are given for charge depletion/accumulation on figure 3. In addition, charge transfer from Ni to Mo₂N is invoked in the manuscript by inspection of the DFT charge density plots – but these densities only show an accumulation in the interlayer region. This is an expected outcome of Ni-surface bonding. It is not necessarily a reflection of electron deficiency in the particles, or accumulation in the substrate. To claim electron density is lost from the cluster and gained by the substrate, total charges projected onto the cluster or surface could be calculated via some standard charge partitioning scheme (such as the Bader method).

Reply: Thanks for review's suggestion. We delete the figure and discussion of charge density in the manuscript.

5) How are the relative energies calculated? NVT AIMD returns internal energies, which will fluctuate significantly about a mean. This mean is likely to vary a lot over the first few ps (and no pre-equilibration is mentioned in this manuscript). A plot of the total energy against time would be useful, along with a description of how the relative energy is extracted.

Reply: Thanks for review's comment. We have added calculation details in the supplementary information and highlighted in blue, the relative energies were calculated by the static DFT calculations converged to local minimum, the initial structures for DFT calculations were from AIMD simulations. The total potential energy against time of Ni₁₉/ γ -Mo₂N(111), Ni₁₉/CeO₂(111), and Ni₁₉O₁₉/O- γ -Mo₂N(111) was plotted in Figure R6. And the total potential energy against time of Ni₄N/ γ -Mo₂N(111) as seen in Figure R10 and was added as Figure S10.

Figure R10. Potential energies of Ni₄N/ γ -Mo₂N(111) during AIMD simulations.

6) The title, and the main conclusion of the paper is that “reverse sintering” is achieved by reduction of Ni particles. But this is not strictly valid. What is shown is a 3d to 2d transition to form layers/islands. No evidence is presented that the Ni particles re-disperse into small clusters or single atoms. Perhaps a different choice of terminology would be more appropriate.

Reply: Thanks for review’s comment. Our experimental results have demonstrated the redispersion phenomenon, it belongs to the concept of reverse sintering, as the large supported particles interacting weaker with the support gradually transform into well-dispersed species strongly contacting the substrate. It can be regarded as the opposite of traditional sintering phenomenon, during which the loaded fine metal clusters escape from the binding of support and fuse into large spherical particles with low interfacial atom ratio and minimized surface energy. As a result, we believe the term of “reverse sintering” is appropriate.

7) It is claimed that the “reverse sintering” increases catalyst longevity, but there is no comparison to base this upon.

Reply: Thanks for review’s comment. The “reverse sintering” increasing catalyst longevity claimed in the manuscript is about the stability of the active site Ni. The support or base Mo_2N is very stable during the reaction, so the long-term stability of Ni/ Mo_2N is mainly depending on the stability of Ni site. Without the strong interaction of Ni and support, the active site may agglomerate to bigger particles leading to the deactivation. Such as the reference of Frei et al (10.1021/jacs.8b11729), the Ni₁/MgO catalyst is not stable when the reaction is up to 350 °C, as the selectivity of CH₄ is appeared and reaches 40% (estimated by the catalytic activity). Even the temperature reduced back to 300 °C, the selectivity of CO cannot remain as good as previous result at 300 °C, indicating the Ni₁ site is not very stable under high reaction temperature.

8) I would appreciate some discussion/speculation of why the selectivity of the catalyst changes upon high temperature reduction. The oxidation state and the physical structure can both play roles, but this is not disentangled in the manuscript.

Reply: Thank for review’s suggestion. We have added some discussion in the manuscript to explain the changes of selectivity of Ni-4nm/ Mo_2N in CO₂ hydrogenation, to relate the catalytic selectivity with the structure of catalyst. “The reduced dimension and electronic deficiency of Ni species are the main reasons for the reduced Ni-4nm/ Mo_2N catalysts active for the RWGS reaction other than methanation.”

9) The referencing is a little lacking. There is a literature on the topic of undercoordinated metals supported on MnX_m (X=N,C) for catalysis, either deposited as layers or dispersed via reduction, e.g. 10.1021/jacs.8b08246 and references within, as well as on the topic of M@oxide SMSI effects. The concept of electronically-induced 3D to 2D transitions of supported metal particles has also been covered extensively, for example by Pacchioni, Freund and colleagues. I would ask the authors to provide a more thorough reference list and put the findings in this context.

Reply: Thanks for the reviewer’s comment. We have studied the suggested papers and other related literature. Those closely related with our topics have been added into the revised references and cited as 15 and 24 and 27.

10) Figure S5 is difficult to interpret. Is the SE intensity increasing between 400 and 500 C? Improved labelling of the figure would help.

Reply: Thanks for the review's comment. We are sorry that we didn't give detailed descriptions about Figure S5. As illustrated in Figure R11, Hitachi HF5000 S/TEM microscopy, which is equipped with ADF, BF and SE detector, can spontaneously record annual dark field (ADF) image, bright field (BF) image and secondary electron (SE) image of an identical object.

As shown in Figure S5, the correlated ADF, BF image give very few information since the sample is too thick for transmission electron microscope, but the SE image gives very clear topographic information about the supported nanoparticles. We corrected this part correspondingly to reduce any misunderstandings.

Figure R11. Schematic illustration of simultaneous acquisition of STEM-Annual Dark field (ADF) image, Bright Field (BF) image and Secondary electron (SE) image in Hitachi HF5000 S/TEM microscopy, which is equipped with ADF, BF and SE detector (copy from reference Ultramicroscopy. 2011 Jun;111(7):865-76)

REVIEWERS' COMMENTS

Reviewer #1 (Remarks to the Author):

I support its publication in Nature Communication. The main concerns of the reviewers have been addressed correctly.

Reviewer #2 (Remarks to the Author):

The reviewers have done a great job addressing my comments. The article can be recommended for publication.

Reviewer #3 (Remarks to the Author):

The authors have a made a good effort to justify and enhance their findings, in particular in the computational results. I am satisfied that the revisions are sufficient.

REVIEWERS' COMMENTS

Reviewer #1 (Remarks to the Author):

I support its publication in Nature Communication. The main concerns of the reviewers have been addressed correctly.

Reviewer #2 (Remarks to the Author):

The reviewers have done a great job addressing my comments. The article can be recommended for publication.

Reviewer #3 (Remarks to the Author):

The authors have a made a good effort to justify and enhance their findings, in particular in the computational results. I am satisfied that the revisions are sufficient.